# Risk Factors for the Development of Nontuberculous Mycobacteria Pulmonary Disease during Long-Term Follow-Up after Lung Cancer Surgery

**DOI:** 10.3390/diagnostics12051086

**Published:** 2022-04-27

**Authors:** Bo-Guen Kim, Yong Soo Choi, Sun Hye Shin, Kyungjong Lee, Sang-Won Um, Hojoong Kim, Jong Ho Cho, Hong Kwan Kim, Jhingook Kim, Young Mog Shim, Byeong-Ho Jeong

**Affiliations:** 1Division of Pulmonary and Critical Care Medicine, Department of Medicine, Samsung Medical Center, School of Medicine, Sungkyunkwan University, Irwon-ro 81, Gangnam-gu, Seoul 06351, Korea; boguen.kim@samsung.com (B.-G.K.); fresh.shin@samsung.com (S.H.S.); kj2011.lee@samsung.com (K.L.); sangwon72.um@samsung.com (S.-W.U.); hj3425.kim@samsung.com (H.K.); 2Department of Thoracic Surgery, Samsung Medical Center, School of Medicine, Sungkyunkwan University, Irwon-ro 81, Gangnam-gu, Seoul 06351, Korea; ysooyah.choi@samsung.com (Y.S.C.); jongho9595.cho@samsung.com (J.H.C.); hkts.kim@samsung.com (H.K.K.); jhingook.kim@samsung.com (J.K.); youngmog.shim@samsung.com (Y.M.S.)

**Keywords:** nontuberculous mycobacterium, lung cancer, surgery

## Abstract

The aim of this study is to determine the cumulative incidence of, and the risk factors for, the development of nontuberculous mycobacteria pulmonary disease (NTM-PD) following lung cancer surgery. We retrospectively analyzed patients with non-small cell lung cancer who underwent surgical resection between 2010 and 2016. Patients who met all the diagnostic criteria in the NTM guidelines were defined as having NTM-PD. Additionally, we classified participants as NTM-positive when NTM were cultured in respiratory specimens, regardless of the diagnostic criteria. We followed 6503 patients for a median of 4.89 years, and NTM-PD and NTM-positive diagnoses occurred in 59 and 156 patients, respectively. The cumulative incidence rates of NTM-PD and NTM-positive were 2.8% and 5.9% at 10 years, respectively. *Mycobacterium avium* complex was the most commonly identified pathogen, and half of the NTM-PD patients had cavitary lesions. Several host-related factors (age > 65 years, body mass index ≤ 18.5 kg/m^2^, interstitial lung disease, bronchiectasis, and bronchiolitis) and treatment-related factors (postoperative pulmonary complications and neoadjuvant/adjuvant treatments) were identified as risk factors for developing NTM-PD and/or being NTM-positive after lung cancer surgery. The incidences of NTM-PD and NTM-positive diagnoses after lung cancer surgery were not low, and half of the NTM-PD patients had cavitary lesions, which are known to progress rapidly and often require treatment. Therefore, it is necessary to raise awareness of NTM-PD development after lung cancer surgery.

## 1. Introduction

Lung cancer remains a major problem because the number of patients is increasing worldwide, and it remains the most common cause of cancer death [1]. However, as low-dose chest computed tomography (CT) is increasingly being used for cancer screening in the general population at risk for lung cancer, the number of patients diagnosed in earlier stages who can undergo surgical treatment has increased [2]. Additionally, clinical staging [3], surgical techniques [4], and neoadjuvant and adjuvant therapies [5,6] have advanced over the last few decades. Although lung cancer treatment is still a challenge, its long-term survival rates should gradually improve with these advances, especially in early-stage lung cancers [7].

Surgical resection is the gold standard treatment for early-stage lung cancer [8]. Unfortunately, patients who develop a postoperative pulmonary complication (PPC) after the surgical resection of lung cancer experience worse long-term outcomes [9]. However, there are insufficient data on the incidence and impact of chronic complications, particularly those associated with chronic pulmonary infections. Nontuberculous mycobacteria (NTM) are ubiquitous environmental organisms that cause chronic pulmonary disease (PD), and the burdens of NTM-PD are increasing globally [10,11]. Furthermore, the incidence and prevalence of NTM-PD rapidly increased in South Korea from 2003 to 2016 [12]. Well-known risk factors for NTM-PD include an older age, underlying structural PD such as chronic obstructive lung disease (COPD), bronchiectasis, interstitial lung disease (ILD), a previous history of pulmonary tuberculosis (TB) [13,14], and the use of immunosuppressant medications [15,16].

Studies on the relationship between chronic pulmonary infection, not NTM-PD, and lung cancer surgery have been reported. Previous studies reported that lung cancer surgery is one of the risk factors for developing chronic pulmonary aspergillosis [17,18]. To the best of our knowledge, few studies have investigated the development of NTM-PD after the long-term follow-up in patients who underwent lung cancer resection surgery. Therefore, we aimed to determine the cumulative incidence of NTM-PD after lung cancer surgery and evaluate the risk factors related to the development of NTM-PD.

## 2. Materials and Methods

### 2.1. Study Population and Data Collection

This was a retrospective cohort study. We screened the medical data of patients with non-small cell lung cancers (NSCLC) who underwent lung resection surgery between January 2010 and December 2016 from the Lung Cancer Surgery Registry at Samsung Medical Center, a 1997 bed referral hospital in South Korea. Patients with a concurrent diagnosis or a previous history of NTM-PD at the time of surgery were excluded. Even when the diagnostic criteria for NTM-PD were not fully satisfied, patients with culture-positive NTM from the respiratory specimen obtained before surgery were also excluded. Additionally, patients who showed granulomatous inflammation in surgical specimens of lung cancer in which NTM infection could not be ruled out were excluded from this study.

We used the same database to gather the following information: patient-related factors, such as age, sex, body mass index (BMI), a history of smoking, underlying pulmonary diseases, other comorbidities, and CT findings at the time of the lung cancer diagnosis (TB sequelae, bronchiectasis, and centrilobular bronchiolitis); cancer-related factors, such as histologic type, location of the tumor, and the clinical/pathological stage of cancer; treatment-related factors, including the neoadjuvant or adjuvant treatments used, the surgical approach, the extent of surgical resection, and the development of any PPCs within 30 days after surgery. Underlying pulmonary diseases included a previous history of pulmonary TB, small airway disease (COPD, asthma), and ILD. The tumor was staged using the Seventh Edition of the American Joint Committee on Cancer [19]. A PPC was defined as the development of any intrathoracic complications during the patient’s hospital stay or during a readmission within 30 days after surgery [20]. Patient follow-up data were last updated in February 2021.

This study obtained approval from the Institutional Review Board (IRB no. 2021-04-016) to review and publish information from patient records, and the requirement for informed consent was waived because the patient information was de-identified and anonymized prior to the retrospective analysis.

### 2.2. Diagnosis of NTM

After surgical resection for NSCLC, most patients were followed-up for at least five years by a thoracic surgeon. Patients with pre-existing or newly developed pulmonary disease were jointly followed-up by a pulmonologist [17]. Physical examinations, laboratory tests, chest radiographies, and chest CT scans were regularly performed at scheduled intervals during follow-up visits. When the development of NTM-PD was suspected based on chest CT images and pulmonary symptoms, patients were referred to a pulmonologist, and further diagnostic tests were performed as needed. The diagnosis of NTM-PD was established as follows: (i) cultured in at least two separate sputum samples or (ii) cultured in at least one or more bronchoalveolar lavage/washing specimens or (iii) lung biopsy with mycobacterial histopathologic features and positive for NTM in tissue culture or in one or more sputum or bronchial lavage fluid cultures [21]. Of course, the final diagnosis was accompanied by an appropriate exclusion of other diseases.

The diagnosis of NTM-PD usually takes months to years even in patients with suspicious clinicoradiological findings [22]. In consideration of this cohort’s many deaths due to lung cancer or loss to follow-up, the following additional situations were also defined: (1) suspicious NTM-PD was defined when the microbiologic criteria were not satisfied, such as only one sputum culture-positive specimen or insufficient identification test results for NTM species; (2) NTM-positive was defined as the sum of all patients with NTM-PD and suspicious NTM-PD.

Acid-fast bacilli (AFB) smears and cultures were prepared using standard methods [23]. All specimens were cultured both on 3% Ogawa solid media (Shinyang, Seoul, South Korea) and in liquid broth media in mycobacterial growth indicator tubes (Becton, Dickinson and Co., Sparks, MD, USA). NTM species were identified using nested multiplex polymerase chain reaction and a reverse-hybridization assay of the internal transcribed spacer region (AdvanSureTM Mycobacteria GenoBlot Assay; LG Life Sciences, Seoul, Korea) [24].

The radiological classification of patients with NTM-positive results was as follows. The fibrocavitary form of NTM-PD was defined by the presence of cavitary opacities predominantly in the upper lobes. The nodular bronchiectatic (NB) type was characterized by the presence of multifocal bronchiectasis and clusters of small nodules [25]. Additionally, the NB form was divided into a “with cavity” and a “without cavity” form.

### 2.3. Statistical Analyses

The data are presented as a number (%) for categorical variables and as the median (interquartile range [IQR]) for continuous variables. Data were compared using the Chi-square test or Fisher’s exact test for categorical variables, and the Mann–Whitney U test for continuous variables. P values for categorical variables with an ordinal scale were calculated with the use of a Mantel–Haenszel test (trend test). The Kaplan–Meier method was used to estimate the cumulative incidence of NTM-PD and overall survival (OS) after the lung cancer surgery.

A multivariable Cox proportional hazard analysis with a backward stepwise selection with *p* < 0.05 for entry and *p* > 0.10 for removal was used to identify the independent risk factors related to NTM-PD development. The clinical stage of the tumor was not included in the multivariable analysis because it had significant collinearity with neoadjuvant and adjuvant treatments. All analyses were performed for both patients with NTM-PD and those who were NTM-positive, respectively. All tests were two-sided, and a *p*-value < 0.05 was considered significant. All statistical analyses were performed using SPSS software (IBM SPSS Statistics ver. 27, Chicago, IL, USA).

## 3. Results

### 3.1. Study Population

Between January 2010 and December 2016, 6789 patients underwent lung cancer surgery, and 6503 patients were finally analyzed (Figure 1). Of the 6503 patients, 156 (2.4%) patients had one or more positive results of NTM culture in their respiratory specimens after lung cancer surgery, and 59 (0.9%) patients were confirmed to have progressed to NTM-PD.

The patients were followed up for a median of 4.89 (IQR: 3.31–6.32) years, and the 5-year survival rate was 76.7% (Figure 2A). During the follow-up period after lung cancer surgery, 59 patients developed NTM-PD at a median of 3.41 (IQR: 2.17–5.08) years (Figure 2B). The cumulative incidences were 0.1%, 0.5%, 0.9%, and 2.8% at 1, 3, 5, and 10 years, respectively, and the incidence rate was 1.9 (95% confidence interval [CI]: 1.5–2.5) per 1000 person-years. Including suspicious NTM-PD, 156 patients had NTM-positive results on their respiratory specimens at a median of 2.79 (IQR: 1.43–4.55) years. The cumulative incidences for NTM-positive results were 0.5%, 1.5%, 2.5%, and 5.9% at 1, 3, 5, and 10 years, respectively, and the incidence rate was 5.1 (95% CI: 4.3–6.0) per 1000 person-years.

### 3.2. Baseline Characteristics of Patients Who Developed NTM-PD and Were NTM-Positive after Lung Cancer Surgery

The median age of the study population was 63 years, and 61.2% were male (Table 1). Patients with NTM-PD had a greater distribution of being > 65 years (59.3% vs. 42.7%, *p* = 0.010), having a smoking history (*p* = 0.007), and having a BMI ≤ 18.5 kg/m^2^ (10.2% vs. 2.6%, *p* = 0.005). In addition, the proportion of patients with bronchiectasis (18.6% vs. 6.1%, *p* = 0.001) and centrilobular bronchiolitis (13.6% vs. 2.3%, *p* < 0.001) on CT images at the time of the lung cancer diagnosis was higher.

Similar trends were observed when the same analysis was performed after dividing patients by their NTM culture results for this cohort (Appendix A). In addition to the trends of the characteristics mentioned above, patients who developed NTM-positive results were more likely to be male (73.7% vs. 60.9%, *p* = 0.001), have a previous history of pulmonary TB (22.4% vs. 10.5%, *p* < 0.001), COPD, or asthma (39.7% vs. 26.7%, *p* < 0.001), and have a higher clinical stage at diagnosis (*p* for trend = 0.001) than those who did not develop the NTM-positive results.

Among the factors associated with lung cancer treatment, patients who developed NTM-PD more commonly received neoadjuvant treatment (20.3% vs. 9.5%, *p* = 0.005) than those who did not (Table 2). In addition, the incidence of PPCs (27.1% vs. 16.8%, *p* = 0.035) was higher in patients with NTM-PD than it was in those without NTM-PD.

Along with these differences, patients with NTM-positive results were also more likely to receive an adjuvant treatment (35.9% vs. 27.9%, *p* = 0.018) than those without NTM-positive results (Appendix A). There were no statistical differences in the extent of surgical resection and pathologic stage whether patients developed NTM-PD or not as well as among patients with or without NTM-positive results.

Among the 59 NTM-PD patients, Mycobacterium avium complex (MAC) was the most common pathogen (Table 3). NTM-PD patients with the NB form accounted for 70%, and the NB form with cavity lesions accounted for 20%. In the 97 patients classified with suspicious NTM-PD, the reasons for not satisfying the microbiologic criteria were as follows: NTM culture (+) from respiratory specimens without an identification test for NTM species (*n* = 68), NTM culture (+) from only one sputum sample with an identification test for NTM species (*n* = 25), and NTM culture (+) from at least two sputum samples with only one identification test for NTM species (*n* = 4). In suspicious NTM-PD patients, about 80% of patients had the NB form, and four (4.1%) of them had cavity lesions (Appendix A).

### 3.3. Factors Associated with the Development of NTM-PD and NTM-Positive Results

The emergence of NTM-PD was independently associated with an age > 65 years (adjusted hazard ratio (aHR): 2.44; 95% CI: 1.43–4.16; *p* = 0.001), a BMI ≤ 18.5 kg/m^2^ (aHR: 3.85; 95% CI: 1.62–9.16; *p* = 0.002), ILD (aHR: 8.23; 95% CI: 1.96–34.51; *p* = 0.004), bronchiectasis (aHR: 2.38; 95% CI: 1.16–4.91; *p* = 0.019) or centrilobular bronchiolitis (aHR: 3.91; 95% CI: 1.71–8.93; *p* = 0.001) on CT imaging at the time of lung cancer diagnosis, PPCs (aHR: 1.90; 95% CI: 1.07–3.39; *p* = 0.029), and treatment with both chemotherapy and radiotherapy (aHR: 2.70; 95% CI: 1.42–5.12; *p* = 0.002) (Table 4).

In contrast, the development of NTM-positive (NTM-PD and suspicious NTM-PD) results was not associated with a lower BMI out of the seven variables, while ever-smokers (aHR, 1.48; 95% CI: 1.02–2.13), a history of pulmonary TB (aHR: 2.27; 95% CI: 1.55–3.31; *p* < 0.001), and thoracostomy (aHR: 1.53; 95% CI: 1.07–2.21; *p* = 0.021) were additionally related (Appendix A).

## 4. Discussion

During the long-term follow-up of 6503 patients undergoing lung cancer surgery, NTM-PD and NTM-positive patients occurred as 2.8% and 5.9% of the 10-year cumulative incidence, respectively. Among the total 59 patients who developed NTM-PD, MAC was the most common pathogen, and the incidence rate of the cavitary disease was 50%. Risk factors related to the development of NTM-PD were an older age, a lower BMI, underlying ILD, bronchiectasis and centrilobular bronchiolitis upon CT imaging, PPCs, and treatment with chemotherapy and radiotherapy. In addition, ever-smokers, a history of pulmonary TB, and thoracotomies were found to be factors that influenced whether a patient was NTM culture-positive.

Although the incidence of NTM-PD varies by region, it has been increasing worldwide, over the past few decades. In a tertiary referral hospital setting in South Korea, one study reported incidence rates of NTM-PD and NTM-positive to be 4.8 and 19.6 per 100,000 person-years in 2016, respectively [26]. In our study, the incidence rates of NTM-PD and NTM-positive results after lung cancer surgery were 1.9 and 5.1 per 1000 person-years, respectively. These incidence rates were approximately 40-fold and 26-fold higher, respectively, than for the general patient population that visits tertiary hospitals in South Korea.

The association between lung cancer surgery and the development of NTM-PD is not well established. There is only one previous study that reported 23 patients with NTM growth in at least one respiratory specimen culture of about 400 patients undergoing lung cancer surgery, and 12 patients met the NTM-PD diagnostic criteria [27]. They analyzed a small number of patients and focused on survival rather than incidence rate, whereas our study provided unprecedented evidence of the NTM-PD incidence after lung cancer surgery by presenting long-term follow-up results using the large-scale cohort data of approximately 7000 patients.

Similar to previous findings that the majority of NTM-PD is caused by MAC [28,29], MAC was the most common NTM species in our study. According to a large study from South Korea, the non-cavitary NB form accounts for about 70% of cases [29]. Compared to this study, we reported that half of the NTM-PD patients had cavity lesions. Considering that cavity lesions in NTM-PD were strongly associated with a poor outcome [29], we suggest that NTM-PD should be intensively monitored in patients after lung cancer surgery.

In our study, an age > 65 years and a BMI ≤ 18.5 kg/m^2^ predisposed patients to the development of NTM-PD. These results are similar to those reported by previous studies in the general population [10,30]. Comorbidities with structural lung disease, such as a previous history of pulmonary TB, bronchiectasis, COPD, and ILD, were also well-known risk factors for developing NTM-PD [15,31,32]. In our study, a previous history of pulmonary TB, ILD, and bronchiectasis were also related to the development of NTM-PD, except COPD. The use of inhaled corticosteroids in COPD patients was a strong risk factor for developing NTM-PD [33]. In our study, most patients diagnosed with COPD were found through preoperative lung function tests, and their lung function was guaranteed to be adequate for surgery. These patients were distinct from those with advanced COPD, who had structural defects and used inhaled corticosteroids. Consequently, COPD should not have emerged as a risk factor in our analysis.

The presence of bronchiectasis or centrilobular bronchiolitis on CT imaging at the time of a lung cancer diagnosis was a related factor for developing NTM-PD in this study. However, some patients with bronchiectasis or centrilobular bronchiolitis on CT images might already have had NTM-PD at that time. A diagnosis of NTM-PD requires repeat tests using respiratory specimens, and the results of these tests need to meet the diagnostic criteria; therefore, NTM-PD detection might be underestimated at the time of a lung cancer diagnosis. Due to our concerns about this very situation, we excluded not only patients who had already been diagnosed with NTM-PD, but also those confirmed to be culture-positive for NTM even once based on respiratory specimens and those with suspicious pathologic results on surgical specimens.

We demonstrated that a PPC was a factor that influenced the development of NTM-PD. This finding suggested that structural defects caused by PPCs might contribute to the development of NTM-PD. Additionally, patients who underwent an open thoracotomy were at a greater risk of developing NTM-PD compared with those who received video-assisted thoracoscopic surgery. We suggest that patients who undergo a thoracotomy are likely to have structural defects present, such as pleural adhesion, which may contribute to the development of NTM-PD after lung resection surgery. Meanwhile, treatment with chemotherapy and radiotherapy increased the risk of NTM-PD, which means that the immunocompromised state that results from chemotherapy and the lung injuries caused by radiotherapy contributed to the development of the NTM-PD [34,35]. To our knowledge, the present study is the first to demonstrate that lung cancer surgery-related factors and PPCs were related to the development of NTM-PD.

This study had several limitations. First, this was a retrospective cohort study of a single institution, which can be a source of selection bias. Second, it is possible that the results were underestimated because a diagnosis of NTM-PD requires repeated AFB cultures and NTM species identifications and usually takes several months to diagnose [22]. Because of this possibility, we included cases suspicious for NTM-PD in the analysis that did not satisfy the microbiological criteria but still reported NTM growth. Previous studies also found that a single NTM growth isolate was associated with the future occurrence of NTM-PD [36]. Despite these limitations, we found that previous risk factors of NTM-PD could be risk factors even in patients who are followed-up for a long time after lung cancer surgery. We also suggest that surgical-related factors and neoadjuvant/adjuvant therapy might influence the development of NTM-PD.

## 5. Conclusions

The cumulative incidence rate of NTM-PD at 10 years after lung cancer surgery was 2.8% (1.9 cases per 1000 person-years), which was approximately 40 times higher than that of the general population in South Korea. Half of NTM-PD patients had a cavitary lesion, which progresses rapidly and often requires treatment. In addition, host-related factors commonly known as risk factors for NTM-PD development were also identified as risk factors in patients with lung cancer surgery. Notably, lung cancer treatment-related factors, such as thoracotomies, PPCs, and additional treatment with chemotherapy and radiotherapy, were associated with the development of NTM-PD or NTM-positive results. Therefore, it is necessary to raise awareness of NTM-PD development after lung cancer surgery, especially in patients with risk factors.

## Figures and Tables

**Figure 1 diagnostics-12-01086-f001:**
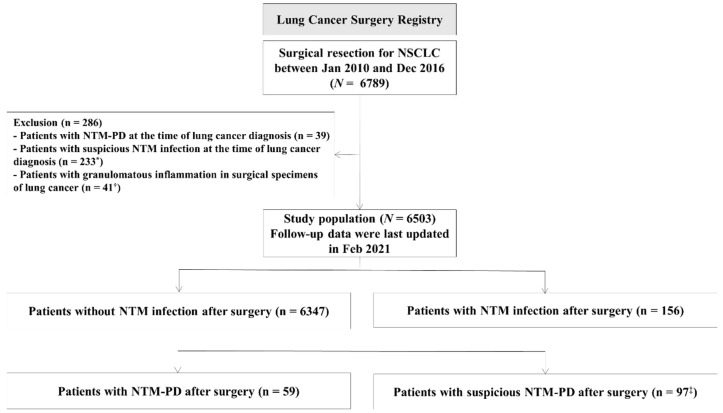
Flow diagram of the study population. * NTM culture (+) from respiratory specimens without an identification test for NTM species (*n* = 44), NTM culture (+) from only one sputum sample with an identification test for NTM species (*n* = 7), NTM culture (+) from only one sputum sample without an identification test for NTM species (*n* = 182). ^†^ Of these 41 patients, 27 patients were included in “Patients with NTM-PD at the time of lung cancer diagnosis (*n* = 16)” or “Patients with suspicious NTM infection at the time of lung cancer diagnosis (*n* = 11)”. ^‡^ NTM culture (+) from respiratory specimens without an identification test for NTM species (*n* = 68), NTM culture (+) from only one sputum sample with an identification test for NTM species (*n* = 25), NTM culture (+) from at least two sputum samples with only one identification test for NTM species (*n* = 4). NTM, nontuberculous mycobacteria.

**Figure 2 diagnostics-12-01086-f002:**
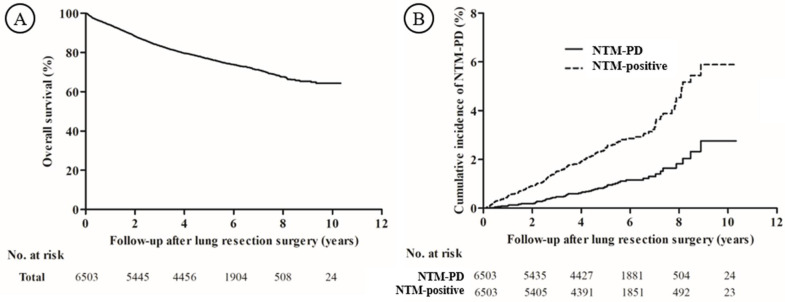
(**A**) The overall survival rate of the study population and (**B**) the cumulative incidence of NTM-PD and NTM-positive results after lung resection surgery. NTM-PD, nontuberculous mycobacterial pulmonary disease.

**Table 1 diagnostics-12-01086-t001:** The baseline characteristics of patients with NSCLC and the development of NTM-PD after lung resection.

Variables	NTM-PD (–) (*n* = 6444)	NTM-PD (+) (*n* = 59)	*p*
Age, years	63 (56–69)	67 (59–69)	0.053
Age > 65 years	2752 (42.7)	35 (59.3)	0.010
Sex, male	3938 (61.1)	41 (69.5)	0.189
Smoking status (*n* = 6501)			0.007
Never smoker	2727 (42.3)	22 (37.3)	
Ex-smoker	2029 (31.5)	11 (18.6)	
Current smoker	1686 (26.2)	26 (44.1)	
Pack-years (*n* = 3761)	30 (20–45)	35 (16–50)	0.694
BMI, kg/m^2^	23.9 (22.0–25.8)	21.9 (20.2–23.8)	<0.001
BMI ≤ 18.5 kg/m^2^	167 (2.6)	6 (10.2)	0.005
Comorbidity			
Pulmonary disease			
History of pulmonary TB	690 (10.7)	11 (18.6)	0.058
COPD/Asthma	1741 (27.0)	18 (30.5)	0.458
Interstitial lung disease	72 (1.1)	2 (3.4)	0.145
DM	1015 (15.8)	6 (10.2)	0.241
Hypertension	2338 (36.3)	25 (42.4)	0.333
Chronic heart disease	447 (6.9)	4 (6.8)	>0.999
Chronic renal disease	89 (1.4)	0 (0.0)	>0.999
Cerebrovascular disease	375 (5.8)	0 (0.0)	0.049
Previous malignancy	890 (13.8)	9 (15.3)	0.749
Clinical stage at diagnosis			0.115 *
Stage I	4450 (69.1)	37 (62.7)	
Stage II	1121 (17.4)	8 (13.6)	
Stage III	812 (12.6)	14(23.7)	
Stage IV	61 (0.9)	0 (0.0)	
Tumor histology			0.516
Adenocarcinoma	4559 (70.7)	43 (72.9)	
Squamous cell carcinoma	1498 (23.2)	11 (18.6)	
Others ^†^	387 (6.0)	5 (8.5)	
Location of lung cancer			0.123
Right	3728 (57.9)	40 (67.8)	
Left	2716 (42.1)	19 (32.2)	
CT findings			
TB sequelae	274 (4.3)	3 (5.1)	0.740
Bronchiectasis	391 (6.1)	11 (18.6)	0.001
Centrilobular bronchiolitis	148 (2.3)	8 (13.6)	<0.001

Data are presented as n (%) or the median (interquartile range). NSCLC, non-small cell lung cancer; NTM-PD, nontuberculous mycobacterial pulmonary disease; BMI, body mass index; TB, tuberculosis; COPD, chronic obstructive pulmonary disease; DM, diabetes mellitus. ** p* values were calculated with the use of a Mantel–Haenszel test (trend test). ^†^ Includes large cell neuroendocrine carcinoma, adenosquamous carcinoma, pleomorphic carcinoma, adenoid cystic carcinoma, mucoepidermoid carcinoma, epithelial myoepithelial carcinoma, and carcinoid tumor.

**Table 2 diagnostics-12-01086-t002:** The treatment profile for NSCLC and the development of NTM-PD after lung resection.

Variables	NTM-PD (–) (*n* = 6444)	NTM-PD (+) (*n* = 59)	*p*
Neoadjuvant treatment			
No	5832 (90.5)	47 (79.7)	0.005
Yes	612 (9.5)	12 (20.3)	
CCRT	531 (8.2)	10 (16.9)	0.028
Chemotherapy	75 (1.2)	1 (1.7)	0.502
Radiotherapy	6 (0.1)	1 (1.7)	0.062
Surgical approach			0.067
VATS	4024 (62.4)	30 (50.8)	
Thoracotomy	2420 (37.6)	29 (49.2)	
Extent of surgical resection			0.238 *
Sublobar resection	1082 (16.8)	6 (10.2)	
Wedge resection	630 (9.8)	4 (6.8)	>0.999
Segmentectomy	452 (7.0)	2 (3.4)	
Lobectomy	4891 (75.9)	50 (84.7)	
Bilobectomy	248 (3.8)	1 (1.7)	
Pneumonectomy	223 (3.5)	2 (3.4)	
Pathologic stage ^†^			>0.999 *
I	4109 (64.4)	36 (63.2)	
II	1193 (18.7)	12 (21.1)	
III	1007 (15.8)	8 (14.0)	
IV	74 (1.2)	1 (1.8)	
PPC ^‡^	1082 (16.8)	16 (27.1)	0.035
Adjuvant treatment ^§^			0.609
No	4636 (72.5)	41 (69.5)	
Yes	1760 (27.5)	18 (30.5)	
CCRT	327 (5.1)	2 (3.4)	0.769
Chemotherapy	1129 (17.5)	11 (18.6)	0.821
Radiotherapy	304 (4.7)	5 (8.5)	0.203

Data are presented as n (%). NSCLC, non-small cell lung cancer; NTM-PD, nontuberculous mycobacteria pulmonary disease; CCRT, concurrent chemoradiotherapy; VATS, video-assisted thoracoscopic surgery; PPC, postoperative pulmonary complication. ** p* values were calculated with the use of a Mantel–Haenszel test (trend test). ^†^ Except for 63 patients where no residual tumor appeared in the surgical specimen after neoadjuvant treatment (pathologic complete response [ypCR]). ^‡^ Pneumothorax and/or prolonged air leak (*n* = 502), respiratory failure that required mechanical ventilation (*n* = 253), pneumonia (*n* = 231), pleural effusion (*n* = 150), others (atelectasis, bronchopleural fistula, pulmonary thromboembolism, etc.) (*n* = 291). Some patients had more than one complication. ^§^ Excluded 48 patients due to data unavailability.

**Table 3 diagnostics-12-01086-t003:** Characteristics of definitive NTM-PD.

Variables	*n* (%)
NTM-PD (*n* = 59)	
Etiology	
*M. avium*	15 (25.4)
*M. intracellulare*	35 (59.3)
*M. massiliense*	2 (3.4)
*M. abscessus*	1 (1.7)
Others *	6 (10.2)
Radiologic findings	
Nodular bronchiectatic form	41 (69.5)
Without cavity	29 (49.2)
With cavity	12 (20.3)
Fibrocavitary form	18 (30.5)

Data are presented as *n* (%). NTM-PD, nontuberculous mycobacteria pulmonary disease. * M. fortuitum complex (*n* = 2), M. kansasii (*n* = 1), M. szulgai (*n* = 1), M. peregrinum (*n* = 1), and M. gordonae (*n* = 1).

**Table 4 diagnostics-12-01086-t004:** Prognostic factors associated with the development of NTM-PD after lung resection for NSCLC (*n* = 6503).

Variables	Univariable Cox		Multivariable Cox	
	Unadjusted HR (95% CI)	*p*	Adjusted HR (95% CI)	*p*
** *Host-related factors* **				
Age > 65 years	2.72 (1.61–4.58)	<0.001	2.44 (1.43–4.16)	0.001
Sex, male	1.75 (1.01–3.05)	0.047		
BMI ≤ 18.5 kg/m^2^	5.60 (2.41–13.04)	<0.001	3.85 (1.62–9.16)	0.002
Smoking history, yes	1.50 (0.88–2.54)	0.134		
Comorbidity				
History of pulmonary TB	1.98 (1.03–3.82)	0.041		
COPD/Asthma	1.35 (0.77–2.35)	0.292		
ILD	7.34 (1.79–30.16)	0.006	8.23 (1.96–34.51)	0.004
Diabetes mellitus	0.70 (0.30–1.64)	0.413		
History of malignancy	1.14 (0.56–2.32)	0.717		
CT findings				
TB sequelae	1.24 (0.39–3.95)	0.721		
Bronchiectasis	3.33 (1.73–6.42)	<0.001	2.38 (1.16–4.91)	0.019
Centrilobular bronchiolitis	6.72 (3.19–14.16)	<0.001	3.91 (1.71–8.93)	0.001
** *Cancer-related factors* **				
Tumor histology				
Adenocarcinoma	Reference			
Squamous cell carcinoma	1.04 (0.53–2.01)	0.915		
Others *	1.79 (0.71–4.51)	0.219		
** *Treatment-related factors* **				
Surgical approach				
VATS	Reference			
Thoracotomy	2.14 (1.28–3.56)	0.004		
Extent of surgical resection				
Lobectomy	Reference			
Sublobar resection	0.53 (0.23–1.24)	0.144		
Bilobectomy	0.48 (0.07–3.45)	0.463		
Pneumonectomy	1.32 (0.32–5.43)	0.700		
PPC ^†^	2.23 (1.26–3.96)	0.006	1.90 (1.07–3.39)	0.029
Neoadjuvant and adjuvant treatment				
No	Reference		Reference	
CTx or RTx alone	1.00 (0.48–2.07)	0.993	1.14 (0.55–2.38)	0.718
CTx and RTx both	2.24 (1.19–4.22)	0.012	2.70 (1.42–5.12)	0.002

NTM-PD, nontuberculous mycobacteria pulmonary disease; NSCLC, non-small cell lung cancer; HR, hazard ratio; CI, confidential interval; BMI, body mass index; TB, tuberculosis; COPD, chronic obstructive pulmonary disease; ILD, interstitial lung disease; CT, computed tomography; VATS, video-assisted thoracoscopic surgery; PPC, postoperative pulmonary complication; CTx, chemotherapy; RTx, radiotherapy. * Includes large cell neuroendocrine carcinoma, adenosquamous carcinoma, pleomorphic carcinoma, adenoid cystic carcinoma, mucoepidermoid carcinoma, epithelial myoepithelial carcinoma, and carcinoid tumor. ^†^ Pneumothorax and/or prolonged air leak (*n* = 502), respiratory failure that required mechanical ventilation (*n* = 253), pneumonia (*n* = 231), pleural effusion (*n* = 150), others (atelectasis, bronchopleural fistula, pulmonary thromboembolism, etc.) (*n* = 291). Some patients had more than one complication.

## Data Availability

Data and material are available on reasonable request.

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
