# Peer review of "Risk Factors for the Development of Nontuberculous Mycobacteria Pulmonary Disease during Long-Term Follow-Up after Lung Cancer Surgery"

_diagnostics, 2022, doi:10.3390/diagnostics12051086_

Round 1
Reviewer 1 Report
The authors studied NTM in a large group of patients with lung cancer who underwent surgical treatment. Although there are no new findings in the study, I am positive because it is a large series and it is a topic that clinicians should keep in mind.
That being said, I do have some minor criticisms.
- First, the abbreviations in the abstract should be stated in clear terms,
- In the results section, as with the others, 3.1. 'Subsection' should have a specific name,
- The first sentence starting with 'çççç' is unnecessary because Figure 1 and the legend are clear enough.
- Figure 2a is unnecessary,
Author Response
Reviewer #1
The authors studied NTM in a large group of patients with lung cancer who underwent surgical treatment. Although there are no new findings in the study, I am positive because it is a large series and it is a topic that clinicians should keep in mind.
That being said, I do have some minor criticisms.
C1. First, the abbreviations in the abstract should be stated in clear terms,
R1. Thank you for your comments. We modified the abbreviations in the abstract.
C2. In the results section, as with the others, 3.1. 'Subsection' should have a specific name,
R2. Thank you for your comments. We corrected.
C3. The first sentence starting with 'çççç' is unnecessary because Figure 1 and the legend are clear enough.
R3. Thank you. As the reviewer said, the description of Figure 1 is clearly stated in the figure legend. Considering that, we modified the first sentence of the Result section to be more concise.
Results (page 3)
Between January 2010 and December 2016, 6,789 patients underwent lung cancer surgery, and 6,503 patients were finally analyzed, excluding 39 patients with NTM-PD at the time of the lung cancer diagnosis, 233 patients with NTM-positive results even once in respiratory specimen cultures, and 41 patients with granulomatous inflammation in sur-gical specimens (Figure 1).
C4. Figure 2a is unnecessary.
R4. Thank you for your comment.
As the reviewer said, Figure 2a is not directly related to the results of our study. However, considering that our study was a single center and retrospective study, we thought it would be good to provide basic information on the survival of lung cancer patients who underwent surgery in our study population.
The survival of our study population was as shown in Figure 2a, and development of NTM-PD and NTM-positive results in our study population were shown in Figure 2b. So, we want to keep Figure 2a.
Reviewer 2 Report
General comments: This study clarified the expected association and problems with the association between lung cancer and mycobacterial infection by reassessment of retrospective cases. The findings presented by this study indicate that factors in lung cancer surgery and medical treatment are clearly associated with exacerbation of mycobacterial lesions, providing follow-up to patient prognosis and complications. Very useful. 
L68-69: "patients who showed granulomatous inflammation in surgical specimens of lung cancer were excluded from this study."
>>Please explane the reason.
L88~: Diagnosis of NTM: All methods are standard methods and are appropriate.
L120~: Statistical analyses :Appropriate statistical methods are used.
L130~: Results:  The data shown in the table is cumbersome, so if possible, why not provide them as supplemental Materials file, separate from the manuscript, except for the parts that are essential for consideration.
L247~: Discussion: No betrayal results were included, but a very important consideration is easy to understand.
L314: First, "this was a retrospective cohort study of a single institution that handles the largest number of lung cancer surgeries in the country, which might limit the generalization of our results."
>>Is it necessary to think like this? Isn't it a sufficient case that suggests the universality of the findings?
Author Response
Reviewer #2
C1. General comments: This study clarified the expected association and problems with the association between lung cancer and mycobacterial infection by reassessment of retrospective cases. The findings presented by this study indicate that factors in lung cancer surgery and medical treatment are clearly associated with exacerbation of mycobacterial lesions, providing follow-up to patient prognosis and complications. Very useful.
R1. Thank you.
C2. L68-69: "patients who showed granulomatous inflammation in surgical specimens of lung cancer were excluded from this study."
>>Please explain the reason.
R2. Thank you for your comment. We excluded patients who showed granulomatous pathology in surgical specimens of lung cancer, because these cases could not be excluded possibility of NTM infection at the time lung cancer diagnosis.
Of course, cases with tuberculosis confirmed by MTB PCR or AFB Stain & Culture using surgical tissue were not excluded from this study. Even after considering all the above cases, only in cases which the NTM infection was not ruled out additionally excluded in our analysis.
We added explanation in Materials and Methods section.
Materials and Methods, (page 2)
Additionally, patients who showed granulomatous inflammation in surgical specimens of lung cancer in which NTM infection could not be ruled out were excluded from this study.
C3. L88~: Diagnosis of NTM: All methods are standard methods and are appropriate.
R3. Thank you.
C4. L120~: Statistical analyses: Appropriate statistical methods are used.
R4. Thank you.
C5. L130~: Results: The data shown in the table is cumbersome, so if possible, why not provide them as supplemental Materials file, separate from the manuscript, except for the parts that are essential for consideration.
R5. Thank you for your comment. With respect to the reviewer’s comment, we separate the results and provide NTM positive related outcomes as supplementary material.
C6. L247~: Discussion: No betrayal results were included, but a very important consideration is easy to understand.
R6. Thank you.
C7. L314: First, "this was a retrospective cohort study of a single institution that handles the largest number of lung cancer surgeries in the country, which might limit the generalization of our results." Is it necessary to think like this? Isn't it a sufficient case that suggests the universality of the findings?
R7. Thank you for your comment. Although our study was conducted on a large number of cases, the disadvantages of a retrospective single center study should be mentioned. We revised that sentence in Discussion section.
Discussion (page 12)
This study had several limitations. First, this was a retrospective cohort study of a single institution, which can be a source of selection bias.
Reviewer 3 Report
Bo-Guen Kim et al reported factors associated with the Development of Nontuberculous Mycobacteria Pulmonary (NTM) Disease after Lung Cancer Surgery. The research question is interesting and the manuscript is well structured. The authors might consider the following comment to improve it.
1-The background of the manuscript should be improved.
By providing the following information:
-The burden of NTM in general worldwide or /and locally
-The factors described in the literature to be the risk factor for developing NTM
-The gaps worldwide or locally
-The rationale of this study
-The specific objective(s)
2-Line 62: We screened patients with non-small cell lung cancer ……………
It should be “we screened patients medical file……….”
3-Line 263-264 add the reference
4-Figure 1: The follow-up period should be mentioned in the figure
5-Table 1: The top heading of the Colum 3(NTM-PD and NTM-positive) Could be deleted.
6-General comment:
The objective of your study was the following:
To determine the cumulative incidence of NTM-PD after lung cancer surgery and evaluate the risk factors related to the development of NTM-PD.
I am not sure to understand why you presented results for NTM-PD and NTM-PD.
Since you described very well in the method session the variable definition of NTM-positive and NTM-PD, you have to choose one as your outcome definition instead of presenting both. In the end, the reader would not be able to follow you from the objective to the result.
Author Response
Reviewer #3
C1. Bo-Guen Kim et al reported factors associated with the Development of Nontuberculous Mycobacteria Pulmonary (NTM) Disease after Lung Cancer Surgery. The research question is interesting and the manuscript is well structured. The authors might consider the following comment to improve it.
R1. Thank you.
C2. The background of the manuscript should be improved.
By providing the following information:
-The burden of NTM in general worldwide or /and locally
-The factors described in the literature to be the risk factor for developing NTM
-The gaps worldwide or locally
-The rationale of this study
-The specific objective(s)
R2. Thank you for your comment. As the reviewer suggested, we revised Introduction section.
R2.1. The burden of NTM in general worldwide or /and locally:
We added the burden of NTM-PD in South Korea. In South Korea, the incidence and prevalence of NTM infection rapidly increased from 2003 to 2016 [1].
Introduction (page 2)
Also, the incidence and prevalence of NTM-PD are rapidly increasing in South Korea from 2003 to 2016 [12].
R2.2. The factors described in the literature to be the risk factor for developing NTM:
This point already described in Introduction section: “Well-known risk factors for NTM-PD include an older age, underlying structural PD such as chronic obstructive lung disease (COPD), bronchiectasis, interstitial lung disease (ILD), and a previous history of pulmonary tuberculosis (TB), and use of immunosup-pressant medications.”
R2.3. The gaps worldwide or locally:
NTM-PD is a disease that shows an increasing trend in recent years both worldwide and in each country [1-6]. Although there are differences in the increasing rate of each region, it is difficult to describe “the gaps worldwide or locally” because the increasing trend is the same globally. In addition, the description of "the gaps worldwide or locally" does not seem to be directly related to the aim of our study, so it is difficult to describe specifically it in the Introduction section.
R2.4. The rationale of this study:
There are studies on the relationship between chronic pulmonary infections, not NTM-PD, and lung cancer surgery. Previous studies reported that lung cancer surgery is one of the risk factors for developing chronic pulmonary aspergillosis [7, 8]. However, to the best of our knowledge, few studies have investigated the development of NTM-PD after long-term follow-up in patients who underwent lung cancer resection surgery.
Introduction (page 2)
Studies on the relationship between chronic pulmonary infection, not NTM-PD, and lung cancer surgery have been reported. Previous studies reported that lung cancer surgery is one of the risk factors for developing chronic pulmonary aspergillosis [17,18].
R2.5. The specific objective(s):
We already specified the objective of our study in last part of Introduction: “We aimed to determine the cumulative incidence of NTM-PD after lung cancer surgery and evaluate the risk factors related to the development of NTM-PD.”
Reference
- Park, S.C., et al., Prevalence, incidence, and mortality of nontuberculous mycobacterial infection in Korea: a nationwide population-based study. BMC Pulm Med, 2019. 19(1): p. 140.
- Adjemian, J., et al., Epidemiology of Nontuberculous Mycobacteriosis. Semin Respir Crit Care Med, 2018. 39(3): p. 325-335.
- Furuuchi, K., et al., Interrelational changes in the epidemiology and clinical features of nontuberculous mycobacterial pulmonary disease and tuberculosis in a referral hospital in Japan. Respir Med, 2019. 152: p. 74-80.
- Huang, J.J., et al., Prevalence of nontuberculous mycobacteria in a tertiary hospital in Beijing, China, January 2013 to December 2018. BMC Microbiol, 2020. 20(1): p. 158.
- Sharma, S.K. and V. Upadhyay, Non-tuberculous mycobacteria: a disease beyond TB and preparedness in India. Expert Rev Respir Med, 2021. 15(7): p. 949-958.
- Wassilew, N., et al., Pulmonary Disease Caused by Non-Tuberculous Mycobacteria. Respiration, 2016. 91(5): p. 386-402.
- Shin, S.H., et al., Incidence and Risk Factors of Chronic Pulmonary Aspergillosis Development during Long-Term Follow-Up after Lung Cancer Surgery. J Fungi (Basel), 2020. 6(4).
- Tamura, A., et al., Chronic pulmonary aspergillosis as a sequel to lobectomy for lung cancer. Interact Cardiovasc Thorac Surg, 2015. 21(5): p. 650-6.
C3. Line 62: We screened patients with non-small cell lung cancer ……………
It should be “we screened patients medical file……….”
R3. We revised the expression.
Materials and Methods (page 2)
We screened the medical data of patients with non-small cell lung cancer (NSCLC) who underwent lung re-section surgery between January 2010 and December 2016 from the Lung Cancer Surgery Registry at Samsung Medical Center, a 1,997-bed referral hospital in South Korea.
C4. Line 263-264 add the reference
R4. Line 263-264 is description sentence, not excerpts from other reference.
From reference No.26 (“Park, Y.; Kim, C.Y.; Park, M.S.; Kim, Y.S.; Chang, J.; Kang, Y.A. Age- and sex-related characteristics of the increasing trend of nontuberculous mycobacteria pulmonary disease in a tertiary hospital in South Korea from 2006 to 2016. Korean J Intern Med 2020, 35, 1424-1431.”) in manuscript, incidence rates of NTM-PD and NTM-positive in South Korea were 4.8 and 19.6 per 100,000 person-years in 2016. In our study, we calculated the incidence rates of NTM-PD and NTM-positive as per 1,000 person-years. These values are 1.9 and 5.1 per 1,000 person-years, respectively. The “40-fold and 26-fold higher” means 190/4.8 (=39.58) and 510/19.6 (=26.02), respectively. Therefore, in this sentence, we do not need the reference. It is just a comparative description between other study and our study.
Discussion (page 11)
“In a tertiary referral hospital setting in South Korea, one study reported incidence rates of NTM-PD and NTM-positive to be 4.8 and 19.6 per 100,000 person-years in 2016, respectively [26]. In our study, the incidence rates of NTM-PD and NTM-positive results after lung cancer surgery were 1.9 and 5.1 per 1,000 person-years, respectively. These incidence rates were approximately 40-fold and 26-fold higher, respectively, than for the general patient population that visits tertiary hospitals in South Korea.
C5. Figure 1: The follow-up period should be mentioned in the figure
R5. Thank you for your comment. As the reviewer suggested, we revised the Figure 1. We marked the last follow-up date in Figure 1.
C6. Table 1: The top heading of the Colum 3(NTM-PD and NTM-positive) Could be deleted.
R6. Thank you for your comment. With respect to the reviewer’s comment, we separate the results and provide NTM positive related outcomes as supplementary material. In the process of separating the results, the table formatting was also modified.
C7. General comment:
The objective of your study was the following:
To determine the cumulative incidence of NTM-PD after lung cancer surgery and evaluate the risk factors related to the development of NTM-PD.
I am not sure to understand why you presented results for NTM-PD and NTM-PD.
Since you described very well in the method session the variable definition of NTM-positive and NTM-PD, you have to choose one as your outcome definition instead of presenting both. In the end, the reader would not be able to follow you from the objective to the result.
R7. We appreciate the reviewer’s comment. With respect to the reviewer’s comment, we separate the results and provide NTM-positive related outcomes as supplementary material. We presented results of NTM-PD mainly in manuscript.
Round 2
Reviewer 3 Report
I am satisfied with the authors' responses.The manuscript has been significantly improved.
I have no other comments.
Author Response
We thank the reviewer for your interest in our paper and for your sincere comments.